# The Role of Stakeholders in the Adoption of Public–Private Partnerships (PPPs) in Municipal Water Infrastructure Projects: A Stakeholder Theory Perspective

Thulani Mandiriza  and David Johannes Fourie *

School of Public Management and Administration, Faculty of Economic and Management Sciences, University of Pretoria, Pretoria 0002, South Africa; mandiriza@gmail.com
* Correspondence: prof.djfourie@up.ac.za; Tel.: +27-12-420-3472

**Abstract:** South Africa receives insufficient rainfall to meet citizens' water needs and this is compounded by deficiencies in infrastructure for water services because of inadequate investment and a lack of maintenance. Municipal public–private partnerships (PPPs) for water infrastructure are rarely utilised for several reasons. Central to this paper is the evaluation of the role played by various stakeholders in influencing the adoption and subsequent approval of municipal water PPP projects. This study critically examined the role of each stakeholder and how other stakeholders perceive their effectiveness during the approval process of water PPP projects. The conceptualisation and implementation of PPPs involves managing both the public and private stakeholders to achieve the desired outcomes. These diverse stakeholders have different values, anchored by the need for rent extraction (profit maximisation motive) or self-interest, as advanced by stakeholder theory. By means of structured interviews, participants highlighted the limitations of each stakeholder and how these contribute to the negative perception of future PPPs. The obtained data were triangulated with secondary sources. The findings confirmed the pursuit of self-interest by various stakeholders, impacting the pace of PPP adoption of municipal water projects.

**Keywords:** public–private partnerships; municipalities; South Africa; stakeholder theory; water infrastructure



## 1. Introduction

This paper seeks to contribute to the evolving debate about the role of stakeholders in municipal public–private partnerships' (PPPs) governance and approval processes. Over 50 per cent of the global population resides in towns and cities where local government or municipalities are responsible for the provision of most public services, including water [1]. Municipal PPPs, given this context, play an important role as a funding mechanism for public infrastructure, especially in developing countries, such as South Africa, which face fiscal/budgetary constraints. In South Africa, for a municipal water PPP project to be implemented, approval is required from the respective municipal council consents, i.e., from the National Treasury and the respective Provincial Treasury, with support from the National Department of Water and Sanitation, ratepayers, and other interest groups, such as labour unions. Private sector partners, such as firms offering technical/engineering skills and financial and legal skills, can collaborate with the public sector. For the PPP arrangement to work, the private sector stakeholders must find their participation rewarding in terms of financial returns (profit maximising motive), while the public sector and other interest groups prioritise the actual service to be delivered.

In this paper, we explored how each stakeholder evaluates the (in)effectiveness of other stakeholders during the approval of municipal PPP projects. Extant studies have focused on the role of each stakeholder in PPP implementation; that is, the private–private and public–public dimensions of the roles of players in executing a PPP project [2]. Other extant

research has focused on the lack of alignment between public and private partners [3], while still others have highlighted the need for frequent engagement during the contracting stage to improve outcomes [4]. Other studies have found that active and early stakeholder involvement while executing the PPP project enhances ownership and subsequent success of the project [5,6]. Another study focused on the development of best practice for PPP implementation in Ghana and revealed the need for PPP practitioners to consult extensively and engage systematically with political opposition parties, among others, during project inception [7]. Minimal research has explored the interplay between stakeholders and how their perspectives of each other shape the PPP approval process. Accordingly, the cited extant studies lack critical evaluation of how each stakeholder rates the effectiveness of other stakeholders during the PPP approval process. We aimed to provide a deeper understanding of this less explored aspect of this issue. By better understanding views of respective stakeholders, PPP practitioners and stakeholders may better coordinate their actions to improve cohesion and facilitate PPP adoption.

The conceptualisation and implementation of PPPs by governments across the world involves managing stakeholders in the public and private sector to enable the achievement of the desired outcomes. Given the involvement of diverse stakeholders motivated by their own self-interest, the pace at which PPP projects are implemented is influenced by the need to balance these interests. A stakeholder is defined as "persons, groups, neighbourhoods, organisations, institutions, societies, and even the natural environment". The nature of the stakeholder depends on what is at "stake" for the actor and whether that "stake" counts [8]. Self-interest motives generally guide the actions of stakeholders in PPP arrangements. Failure to manage stakeholders has been highlighted either as a macro-level risk associated with political opposition of the project risk, or as a micro-level risk arising from lack of commitment among partners [9].

PPPs are "sophisticated interfaces between public authorities and private sector undertakings with an objective of delivering infrastructure projects, public goods, and services" [10]. PPPs therefore incorporate private actors in the delivery of public goods and services beyond arms-length transactions [11]. The government and private actors' alliance is regarded as essential in PPPs, given the reconfiguration of the traditional government role to that of a collaborator [12]. Self-interest motives from diverse stakeholders often result in ideological and executive tensions, given that PPPs are complex [13,14]. Stakeholders operate at different levels of the PPP value chain: some are policy makers and some are implementers, while others are purely financiers.

Over and above cohesion among stakeholders, public–private partnerships (PPPs) require an enabling environment to facilitate private sector participation. An enabling environment includes the observance of sound governance principles; enforcing the rule of law; existence of credible legal and regulatory institutions; market conditions that promote competitive outcomes; and expert knowledge to prepare PPP documents and deliver complex projects [15].

The article is structured as follows. Firstly, we provide a background of municipal PPPs in the water sector in South Africa and the extent of water infrastructure backlogs. Secondly, we present the theoretical framework anchoring the study, which is stakeholder theory. We next provide a discussion of the actors involved in the municipal PPP approval process and their roles as stipulated in legislation. This is followed by an exposition of the methodology used in the study, and then the findings. We then conclude and highlight some of the contributions of this paper to the body of knowledge.

## 2. Context of Municipal PPPs in South Africa

In this paper, to reflect the role of stakeholders in PPPs, we used the South African experience in relation to municipal water infrastructure projects. We briefly discuss the municipal legislative environment, and the extent of the water infrastructure backlogs in South Africa. This is to show how PPPs may be a credible financing option for municipalities, yet are underutilised, partly due to the influence of stakeholders.

The municipal environment in South Africa is governed by various pieces of legislation, of which the following have relevance to PPPs: Local Government: Municipal Structures Act, 1998 (Act 117 of 1998), Local Government: Municipal Finance Management Act, 2003 (Act 56 of 2003) (MFMA), and the Municipal PPP Regulations of 2005. The legislation above identifies the stakeholders that play a role in the PPP approval process and how the stakeholders should interact. The applicable stakeholders are relevant sector government departments, organised labour, and the public/ratepayers (this is expanded upon later in this paper).

South Africa experiences rainfall shortages, deficient water for consumption, and inadequate water infrastructure [16], yet the use of PPPs in the water sector remains limited. In South Africa, municipalities (local government) are the designated water services authorities responsible for delivering water to residents and businesses. There are 257 municipalities in South Africa, of which only 144 are designated as water services authorities [17]. Municipalities not designated to provide water services may appoint other municipalities to provide water or utilise water boards for an agreed fee, or, alternatively, utilise private sector providers via a concessionaire or management/lease agreement [18]. The infrastructure of most water service authorities is in dire state; for instance, at least 33 per cent of them are deemed dysfunctional and more than 50 per cent of the municipalities have technical skill deficiencies [19]. PPPs may resolve some of these challenges, as the private sector brings expert technical knowledge, a large pool of financial resources, and project management capability [20].

South Africa's Department of Water and Sanitation (DWS) has estimated that ZAR 840 billion is needed to fund water infrastructure, inclusive of maintenance costs, in the national and local government spheres, with a funding shortfall of approximately ZAR 333 billion for the upcoming ten-year period [19]. The main options that municipalities can pursue to finance water infrastructure include national government grants, raising capital from financial institutions, municipal raised revenue (from rates, taxes, levies), and PPPs [21]. South Africa's infrastructure investment via PPPs is below two per cent of public infrastructure investment, whereas the United Kingdom has achieved almost 50 per cent [22]. South Africa lags behind most developing countries, which are increasingly using private capital to fund public infrastructure [23]. South Africa's municipalities are not utilising water infrastructure PPPs extensively and the only notable examples are the following:

- Queenstown Local Municipality (now Lukhanji Local Municipality) in 1992;
- Stutterheim Local Municipality (now Amahlati Local Municipality) in 1993,
- Fort Beaufort Local Municipality (now Raymond Mhlaba Local Municipality) in 1995;
- Nelspruit Local Municipality (now City of Mbombela Municipality) in 1999;
- Dolphin Coast (now KwaDukuza) in 1999 [24].

Since the municipal PPP guidelines came into effect in 2005, no concession agreements have materialised in the municipal water sector, and this paper sought to explore whether the involvement of multiple stakeholders in the PPP approval process has contributed to the status quo.

## 3. Theoretical Framework

Stakeholder theory is premised on the interference between critical actors within or outside of an organisation [25]. A stakeholder is "any group or individual who can affect or is affected by the achievement of the organisation's objectives" [26]. The literature on the definition of a "stakeholder" has identified several factors; for instance, the nature and level of one's participation in the project, relationship with the project, nature of the claim in the project, a party's role in the project, and the anticipated behaviour of an actor towards the project [27]. Stakeholders are therefore direct or indirect representatives with some contribution to make to a specific project. The representatives may have some material or legitimate claim to the success or failure of the proposed project. Stakeholders may also include the government's regulatory agencies, who act as gatekeepers [27]. The

theoretical underpinnings of stakeholder management in projects are predicated on fairness, social rights, and equality [28]. Stakeholder theory is therefore founded on the notion that organisations ought to be governed in a way that provides mutual benefits to all actors, such as equity shareholders, workers, customers, contractors and the general public, the government as regulators, and the government as service providers [27].

The diverse interests of actors promote the pursuit of self-interest, which inherently leads to conflict given the complex nature of most PPP projects [13,14]. In the context of municipal PPPs in South Africa, stakeholders include those in policy formulation (national government), implementers (local governments), and delivery of services (private sector institutions). In addition, other relevant stakeholders include political parties (represented in councils), special interest lobbyists, residents, representatives of businesses, and labour unions [29]. In PPP arrangements, the primary goal is to foster private and public collaboration and share the associated risks and rewards, while delivering agreed projects [27]. Even in instances where there is a perceived alignment of project goals, stakeholders are still guided by the need to enhance their own incentives (economic rent), and similarly government regulators seek to minimise overall project risks from a financial perspective. Extant studies have highlighted the positive relationship between stakeholder management and achievement of organisational outcomes [30], while others have focused on the relationships of stakeholders during the implementation of PPP projects [31]. The role of stakeholders in the successful implementation of PPPs and factors affecting the operational performance have also received much attention [31]. Earlier research analysed how stakeholders are included in PPP projects, and it explored ways of improving their engagement [32]. As alluded to in the introductory section, there have been limited studies that focus on the actual interplay of stakeholders during the approval process of PPP projects, and this study seeks to make a contribution in this area.

Stakeholders manage their relationships either formally or informally in an inter-organisational governance system. The rules of engagement between partners, either formal or informal, reflect the inter-organisational governance system. Inter-organisational governance mechanisms can be contractual, with formalised and legally binding contracts, or relational, anchored in informal social interactions [33]. These inter-organisational governance systems complement one another, and they can also be substitutes for each other in some instances [34].

In addition to complex stakeholder management, PPPs involve the bundling of services through the use of specialised skills in a single project. This bundling of services from various actors adds to PPP complexity, as these services create vertical linkages among stakeholders, especially for large PPP projects [35]. Bundling involves the provision of "long-term integrated solutions consisting of bundles of interrelated goods and services" [36]. The notion of bundling of interrelated services starts from the conceptualisation of the PPP project; for example, from the design, build and operate option, to actual implementation and monitoring of the project. The public sector is expected to develop contract management expertise to enable it to transfer substantial risk to the solution provider over the duration of the PPP arrangement, and this tends to be a key challenge [36]. The private sector, as a solution provider in the context of a PPP arrangement, is expected to co-create and deliver a specialised service while working in partnership with its client (public sector), which has proven to be a challenging endeavour in most cases [37]. Bundling of services is meant to minimise the life cycle costs of the infrastructure assets due to improved economies of scale and scope and enhanced innovation, which ultimately derive considerable benefits to both the public and private partners [38].

Figure 1 shows the vertical relationships among partners in a PPP structure.

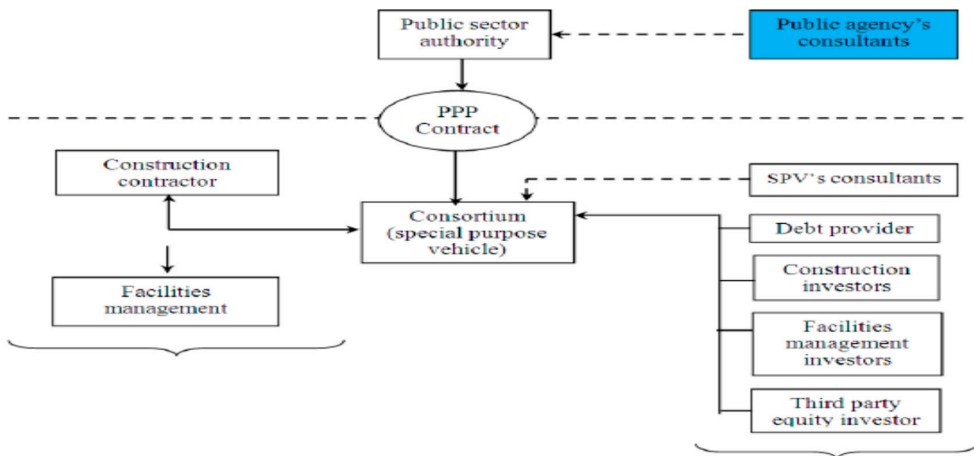

**Figure 1.** Vertical linkages in an illustrative PPP structure. Source: Adapted from [39].

A PPP structure in essence resembles some form of inter-organisational arrangement among stakeholders, governed by an enforceable contract. The responsibilities and rights of each stakeholder are specified, including expected outcomes, and reflected as a performance-based contract [40].

Stakeholder values may manifest as the pursuance of self-interest, which exacerbates PPP complexity, as hypothesised in stakeholder theory [14]. PPPs represent a cooperative outcome for participating stakeholders, but this does not trivialise their distinct values. Figure 2 illustrates the tension between stakeholder values and how bids are evaluated. By developing the bid criteria, the government or the PPP regulator is signalling its own values, which it will use to evaluate the bids [14]. Prior studies on values attribute the government as pursuing public policy objectives which are meant to mitigate negative externalities, harness complementarity of resources, and leverage cost advantages between the public and private entity to derive efficiencies in service delivery [38]. The stakeholder values are represented by $V_1, V_2, \ldots, V_n$ (Figure 2), showing diversity in values, and these values are subject to evaluation criteria by the PPP regulator or sponsor. The preferred bid from a number of bids represented by Bid 1 to Bid 4 is selected on the assumption that there is some degree of alignment of values with the PPP regulator. The assessment criteria used by the PPP regulator are represented by numbers $1, 2, \ldots, 10$ with weights assigned to each criteria (Figure 2).

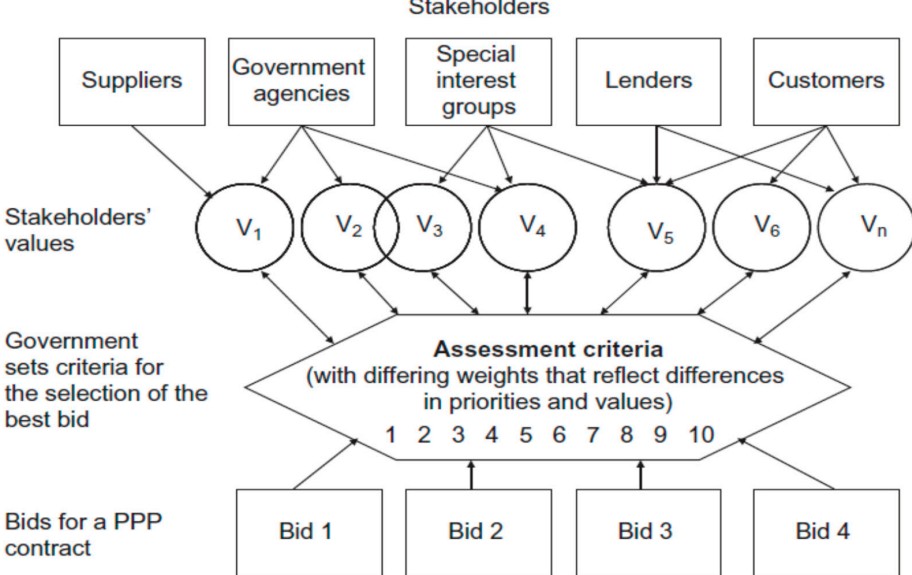

**Figure 2.** Expression of values by PPP stakeholders. Source: Adapted from [14].

As discussed in the introductory section, the key theoretical contribution of this paper is the exposition of how each stakeholder evaluates the (in)effectiveness of other stakeholders during the approval of municipal PPP projects. Extant studies have focused on the role of each stakeholder in PPP implementation [2], alignment at PPP contracting [3], and active and early stakeholder involvement [5,6]. The cited extant studies lack critical evaluation of how each stakeholder perceives the others during the PPP approval process. Clarity on the views of the respective stakeholders is valuable for PPP practitioners to mitigate potential pitfalls and enhance stakeholder cohesion for future PPP projects.

## 4. Actors in the Municipal PPP Approval Process in South Africa

*Life Cycle of Municipal PPP Projects*

The regulations governing municipal PPP projects set out the four main phases which a proposed partnership must follow: "inception, feasibility study, procurement, and PPP contract management". The respective roles of players at each stage of the PPP approval process (in respect of municipal water infrastructure projects) are reflected in Table 1.

**Table 1.** Actors in the municipal PPP approval process.

| PPP Stage | Stakeholder and Role in Water PPPs | Activity |
|---|---|---|
| **Inception phase** | Municipal officials (represented by the accounting officers) | • Initiation of the project by clearly defining the scope of the project requiring the PPP.<br>• Municipal officials appoint an internal project officer or utilise an external adviser.<br>• Mandatory notification to the National Treasury, respective Provincial Treasury, national department responsible for the specific service requiring the PPP (Department of Water and Sanitation), and Department of Cooperative Governance and Traditional Affairs.<br>• Public awareness of the scope of the proposed PPP for comments. |
| | Municipal council | • Approval of the project plans related to the PPP, inclusive of the budget for the project. |
| | National Treasury | • Receive notification or registration of the PPP.<br>• Liaise with the National Treasury.<br>• Determine if the proposed project requires a feasibility study. |
| | Provincial Treasury | • Receive notification or registration of the PPP. |
| | Department of Water and Sanitation | • Receive notification or registration of the PPP. |
| | Department of Cooperative Governance and Traditional Affairs | • Receive notification or registration of the PPP. |
| | Public (rate payers) | • Solicit views from the residents through a public consultation process. |
| **Feasibility study** | Municipal officials (represented by the accounting officers) | • Needs analysis and technical evaluation of funding/delivery options.<br>• Develop a feasibility study report for consideration by the Council and other stakeholders.<br>• Obtain views of organised labour and the public. |
| | Municipal council | • In-principle council approval of the feasibility study and related project plans. |
| | National Treasury | • Provide comments, views, and recommendations on the PPP project (TVR I). |
| | Provincial Treasury | • Issue comments, views, and recommendations on the PPP project. |
| | Department of Water and Sanitation | • Provide comments, views, and recommendations on the proposed PPP project. |
| | Department of Cooperative Governance and Traditional Affairs | • Provide comments, views, and recommendations on the proposed PPP project. |
| | Independent transaction advisors (private sector) | • Assist municipal officials to develop the feasibility study. |
| | Public (rate payers) | • Provide views through a public consultation process. |

**Table 1.** *Cont.*

| PPP Stage | Stakeholder and Role in Water PPPs | Activity |
|---|---|---|
| Procurement—competitive bidding process | Municipal officials | • Accept bids and assess based on the feasibility study.<br>• Adjudicate and decide on the preferred bidder.<br>• Draft a value assessment report.<br>• Prepare documents for the council's consideration (draft agreements, procurement documents, etc.).<br>• Solicit the National Treasury's comments and recommendations (TVR IIA) on draft agreements.<br>• Solicit the National Treasury's comments and recommendations (TVR IIB) during the negotiation process with the preferred bidder.<br>• Finalise the PPP contract management plan.<br>• Invite further comments from stakeholders, including the National Treasury, on the final PPP agreement (National Treasury's views and recommendations (TVR III). |
| | Municipal council | • Approve the draft of the PPP agreement for implementation.<br>• Authorise the execution of the PPP contract. |
| | National Treasury | • Provide the National Treasury's views and recommendations on the drafted PPP agreement (TVRII A), negotiate with the preferred bidder (TVR IIB), and approve of the final PPP agreement (TVR III). |
| | Private sector consortium (financiers, engineering, project management, etc.) | • Participate in the bidding process. |
| Contract management | Municipal officials | • Accounting officer manages the PPP contract and its implementation. |
| | National Treasury | • Authorise the final PPP agreement. |
| | Municipal council | • Monitor performance of the PPP agreement. |
| | Private sector consortium | • Implement the PPP project. |

Authors formulation based on [41].

As illustrated in Table 1, there are various stakeholders in the PPP approval process from all spheres of government, as well as relevant sector departments, organised labour, and the public/ratepayers. As indicated previously, self-interest impacts how the stakeholders interact with one another, given their respective roles [14].

• The National Treasury's role is to ensure that the municipality can afford the PPP arrangements.
• The Provincial Treasury's role is similar to that of the National Treasury, but it also brings its knowledge of the local dynamics.
• The Department of Water and Sanitation ensures water-related projects are carried out in compliance with set standards and applicable laws. In addition, the views of the DWS as a water regulator enhance the overall project credibility.
• The Municipal Council approves PPP projects and the associated funding, and it provides oversight of the implementation of the project.
• The private sector consortium comprises engineering, financial, legal, and other relevant skills. Its incentive is to achieve project objectives at a reasonable return on investment.

Stakeholder theory suggests that when facing disputes, stakeholders converge to find a solution through some overlap in shared rules and norms [26]. Stakeholder conflicts delay projects, increase financing costs, and in some instances lead to project cancellation. "Stakeholder opposition risk" has been identified as a material risk factor for PPPs [42].

## 5. Materials and Methods

This paper was anchored on a previous qualitative case study by the same authors [43], where we assessed factors affecting the pace of adoption of municipal PPP projects in the water sector from a selected sample of Gauteng municipalities in South Africa. The present paper benefited from the interviews conducted for the prior study. A qualitative

case study research methodology was used, and four Gauteng municipalities (City of Johannesburg Metropolitan Municipality, City of Tshwane Metropolitan Municipality, City of Ekurhuleni Metropolitan Municipality, and Midvaal Local Municipality) were selected based on convenience or purposive sampling. Gauteng Province is the economic hub of South Africa, accounting for over one-third of the total gross domestic product [44], and the findings of the study are therefore significant. The three metropolitan municipalities in Gauteng Province account for over 80 per cent of Gauteng's households [45], and this informed the inclusion of all three metros in the case study. Midvaal municipality was chosen as it was the only municipal council under the opposition party (Democratic Alliance) in Gauteng Province at the time the study was undertaken. The preference for a case study approach was based on its flexibility and alignment with the research question [46]. This enhanced the overall interpretation of an occurrence within a social setting [47].

Interviews were conducted with the most relevant officials from the municipal sample, private sector, national government, and provincial government who play a role in municipal water infrastructure financing. The interviewees were selected using nonprobability sampling, as we identified the participants that would form part of the sample based on their role and expertise upfront. In addition to purposive sampling, a snowball technique (chain-referral sampling) was used, and it proved to be effective in identifying potential participants in the public sector where there was no obvious contact list of officials responsible for aspects the researchers were interested in [48]. The interviews were conducted through Microsoft Teams and were recorded and subsequently transcribed to aid the analysis. Thematic analysis was used to process the data gathered, which involved checking patterns in the data and grouping common topics/responses together. Thematic analysis is "a method for identifying, analysing, and interpreting patterns of meaning within qualitative data" [49].

An additional tool, document analysis, complemented the responses from the interviewees, and the secondary data included annual reports from the sampled municipalities and government departments, press releases, and reports from multilateral institutions such as the World Bank, among others. Qualitative content analysis or documentary review analyses the text data from various secondary sources [50]. A total sample of 31 middle to senior management officials participated in the interviews. The participants were as follows:

- Thirteen participants came from the four municipalities;
- Seven participants were from the private sector (financiers, water sector specialists, and PPP experts);
- Eleven participants were from national and provincial government departments (the National Treasury, the Gauteng Provincial Treasury, the Department of Water and Sanitation, and the Department of Cooperative Governance and Traditional Affairs).

To maintain anonymity, names or their respective organisations are not mentioned when reporting the findings. The variation in the respondent groups allowed the researchers to triangulate the responses and improve data robustness [51,52].

## 6. Findings

The evaluation approach adopted to assess the role of each stakeholder was based on what the legislation requires of each stakeholder compared to what is done in practice, and whether other stakeholders see their role as effective or not.

Unsurprisingly, each actor had different views about the role played by other stakeholders during the approval process of PPPs. More interestingly, stakeholders highlighted the inadequacies of various actors in the PPP approval process which impact the adoption of municipal water PPP projects. In the following sections, we summarise the views of participants in relation to each of the stakeholders in the PPP value chain process.

*6.1. National Treasury*

The Constitution of South Africa, in section 216, establishes the National Treasury as the gatekeeper of all financial matters across government. Its responsibilities include developing financial norms, rules, and standards that the whole government should abide by. In addition, the National Treasury monitors expenditure based on allocated budgets. The involvement of the National Treasury in municipal PPPs is guided by the Local Government: Municipal Finance Management Act (Act 53 of 2003) (MFMA). The MFMA authorises the National Treasury "to monitor municipal budgets, monitor implementation of approved budgets, examine borrowing trends, advise on budget compilation, issue guidelines for budget preparation, and carry out in-year budget monitoring and financial oversight to municipalities, among others".

Stakeholders questioned the considerable role played by the National Treasury in the approval process of PPPs and equated their role to that of a "stubborn" gatekeeper. While most stakeholders (94 per cent, or twenty-nine out of thirty-one stakeholders) identified the need for the National Treasury to assist in the value-for-money assessment, the extent of their involvement from feasibility to project closure was deemed to be excessive and to some extent an overreach. Seven participants from the private sector (independent PPP experts and financial institutions) and thirteen participants from municipalities blamed the slow pace of PPP adoption on the length of time taken by the National Treasury in granting approvals or recommendations. These participants highlighted that the National Treasury's views and recommendations (TVRs) are taken as "law" or "binding" by municipalities. Four TVR stages are required before a final go-ahead is given. The National Treasury ratifies each stage of the process, yet there are no defined timeframes for this function. As discussed above, the National Treasury issues four TVRs for each application during the PPP cycle, as summarised below:

- TVR I deals with the feasibility study.

Municipalities develop a feasibility study, which is then submitted to the PPP regulator (National Treasury's Budget Office). Once a feasibility study is submitted, the PPP regulator convenes a review committee (Treasury Review Committee) to evaluate the application. The Treasury Review Committee includes other divisions in the National Treasury, such as Intergovernmental Relations, Asset and Liability Management, Public Finance, and the Government Technical Advisory Centre (GTAC).

- TVR IIA involves the ratification of the bid documents and drafted PPP agreement.
- TVR IIB gives the go-ahead for the municipality to negotiate with the preferred bidder and finalise the PPP contract management.
- TVR III authorises the final PPP agreement to be considered and approved by the municipal council and subsequently signed by the accounting officer.

The TVR process is thus repeated four times before a final go-ahead is given. The National Treasury ratifies each stage of the process without defined timeframes to issue its views and recommendations.

All thirteen participants from the municipalities and seven participants from the private sector cited delays in TVRs as a hindrance to the speedy evaluation of PPPs, given the silo mentality within various units of the National Treasury. In an earlier study, we interviewed both the municipalities and the private sector players, who cited delays in TVRs as an obstacle for rapid evaluation of PPPs. To illustrate the time delays, one participant with extensive experience in PPPs from both the private and public sectors summarised the problem as follows:

> *"...if you submit a feasibility study report to the National Treasury today, you will get a response after 6 months, ...they actually delay the whole PPPs process, and that's when politicians come back and say, "I'm tired of PPP; it's not giving me anything; I'm not going to be voted in again—I will not even try"* [53].

Several factors explained the delays, including limited human resources and a lack of personnel focusing exclusively on PPPs. The participants revealed that only three to four National Treasury staff members deal with the entire PPP portfolio for the whole country, and these officials have additional responsibilities aligned to the national budget process [43].

Some participants from the National Treasury acknowledged its role in causing delays in approving PPP projects, but equally noted the limitations of other stakeholders in the value chain; for instance, the lack of skills and capacity at the local government level to develop credible feasibility studies and adequate justification of PPP as a viable financing mechanism. The National Treasury's views were confirmed by participants from the municipalities, as only 23 per cent (3 out of 13) received some basic training on general municipal PPP regulations. One participant noted the following:

*"I was involved in evaluating another PPP tender. So, I have a bit of an insight on the process, but I wouldn't say I've had any proper training, and this is a gap you will find in most municipalities. Most of us don't understand the PPP process, including senior people"* [53].

### 6.2. Government Technical Advisory Centre (GTAC) (An Entity of the National Treasury)

The Budget Office of the National Treasury has housed the PPP Unit since 2000, with the intention of giving special focus to PPPs. The PPP Unit was later transformed into the GTAC, with an extended mandate. The GTAC's role includes "transaction advisory services for PPPs, capital projects appraisal, technical consulting services, public expenditure and policy analysis, jobs fund project management, and the municipal finance improvement programme" [54]. The GTAC combined the "National Treasury's advisory and support activities within a dedicated centre with skilled expertise" [54]. The GTAC also provides technical input/comments to the National Treasury's TVR process, as it is part of the Treasury Review Committee.

Similar to the National Treasury, the GTAC has limited human resources to fully support all PPP activities in South Africa. The Financial Fiscal Commission (FFC) highlighted that "the GTAC lacks the resources to promote PPPs and build capacity within municipalities to originate, implement, and manage the PPPs" [55]. The Technical and Advisory Services Programme, which supports PPPs and other large infrastructure, has 38 staff members [54] to service the entire public sector. The PPP Unit underestimates the number of PPP projects that it plans in a year and is therefore unable to service some of the projects. For instance, in 2018/19, 33 requests for PPP support were made, in comparison with 10 planned projects [54]. It is therefore clear that it is not only about having key stakeholders in the PPP approval process, but the capacity of stakeholders such as the GTAC is important to improve the adoption of PPPs.

Like the National Treasury, participants from GTAC equally blamed the lack of capacity of municipal officials to properly conceptualise and implement PPP projects. Despite the training provided by the GTAC, some municipal officials have not been able to grapple with the complexity of PPP arrangements given their limited technical training. Participants from the GTAC highlighted the exclusive reliance by municipalities on private sector consultants to develop feasibility studies [53].

### 6.3. Municipal Officials and Municipal Councils

Section 46 of the Local Government: Municipal Finance Management Act (Act 53 of 2003) makes provisions for municipalities to borrow to finance capital projects or refinance existing loans [56]. Decisions on which infrastructure financing option a municipality chooses are subject to a political process. Councillors are political representatives who make decisions on budgets, including financing of infrastructure projects such as PPPs.

Municipal councils in South Africa have a term of five years. Participants cited that councillors decide on infrastructure investments linked to their term of office and are therefore reluctant to support PPP projects that take a long time to accomplish. Participants

highlighted that councillors have a desire to leave a visible legacy, which is not possible under a PPP as the project may be completed in a subsequent political term (which they might not be part of). The participants from municipalities were unequivocal that politicians prefer ad hoc projects with immediate visibility once they commence their political term, and therefore do not support PPPs even if they are a viable option for financing water infrastructure projects.

Two participants each from the municipalities and the private sector highlighted that politicians are sensitive to the use of the private sector to deliver public services given the historic race tensions in South Africa. The private sector in South Africa is dominated by White men, and the perception created through PPPs is that the public sector (largely controlled by Black men) has failed. There was consensus among participants that if councillors (elected political leaders) are not supportive of PPPs, then no projects will be financed through PPPs as the municipal council has the power to decide on funding options. A limitation of this study was that no politicians were interviewed, and therefore future studies should focus on the perspective of elected officials on municipal PPPs.

### 6.4. Labour Unions

Section 78(1)(a)(v) of the Municipal Structures Act (Act No. 32 of 2000) requires a municipality to solicit the views of organised labour before PPP projects are approved by the council. PPPs, by their very nature, involve collaboration with the private sector, and this raises concern from public sector labour unions in relation to potential job losses or changes in their conditions of service. Most of the participants mentioned what transpired in the City of Mbombela Municipality, where a PPP water project was delayed because of ferocious opposition from organised labour, who feared being replaced with private sector jobs [57]. The impasse was subsequently resolved when sufficient guarantees were made to the labour unions, but the project was delayed. Most participants from municipalities highlighted that the Mbombela experience has influenced politicians not to consider PPPs as an attractive option in their own municipalities.

Participants cited the alliance between the ruling African National Congress (ANC) and the Congress of South African Trade Unions (COSATU) as contributing to the general hesitancy by ANC politicians to appear to undermine the labour unions by considering PPPs in the water sector. The lack of support from labour unions results in councillors hesitant to push ahead with PPP projects, even if they are a credible delivery mechanism. Water infrastructure projects that are suitable for PPPs are shunned for political reasons, and the need to maintain the cordial relationship between politicians and the labour unions. A limitation of this study was that no labour unions were interviewed; this could be addressed in future studies.

### 6.5. Private Sector

The private sector participants had a different perspective on the influence of other stakeholders when PPP projects are being evaluated. For instance, an international PPP expert indicated that South Africa's municipal PPP framework is aligned to best practice but suffers from poor execution from public sector stakeholders. The public sector stakeholders take time to produce a credible feasibility study (municipal officials) and assessment thereof (National Treasury). Participants from financial institutions assigned some responsibility for the delays in the execution of PPP projects to the unregulated time spent by the National Treasury on the approval process, and the lack of human resources to evaluate an escalating number of projects intended to be financed through PPPs.

Participants from municipalities also highlighted that the private sector makes infrastructure investment more expensive and out of reach of most municipalities. Over 75 per cent of the participants from municipalities cited the high costs associated with PPPs as disincentivising their rapid adoption, given the higher returns required by private sector players.

### 6.6. Department of Water and Sanitation

Water resources management in South Africa falls under the overall authority of the Department of Water and Sanitation (DWS). The DWS oversees all water-related legislation and the associated policies, which cover the entire value-chain, in addition to being a water sector regulator [58]. Most participants highlighted that the role that the DWS plays in the PPP approval process is minimal, but it assists in dealing with any regulatory issues that arise in the implementation of water projects.

Participants also highlighted that the governance structure is fragmented across the water sector value chain, with responsibilities scattered across institutions. Water sector policy and sector regulation are dual roles undertaken by the DWS. This has been cited as blurring water sector transparency and accountability, and thereby disincentivising investment through PPPs. In addition, municipal water services are subject to a trilogy of institutions and associated legislation: the National Treasury on finance-related issues, the DCOG on overall governance matters, and the DWS on the water sector [59]. These stakeholders in the governance framework pursue different objectives, which may slow the pace of PPP adoption in the water sector.

Like the National Treasury and GTAC, participants from the DWS noted the lack of skills at the municipal level as a contributing factor to the limited adoption of PPPs. The DWS noted that municipalities cannot actively pursue PPPs, of which they have limited understanding.

### 6.7. Department of Cooperative Governance (DCOG)

Municipalities in South Africa are accountable to the DCOG. The DCOG's mandate "is to build an efficient and progressive local government system that meets its obligations, as enshrined in the Constitution" [60]. The DCOG coordinates with other spheres of government and other stakeholders to achieve its mandate. The DCOG supports municipalities with capacity for building initiatives, and simultaneously exercises oversight of their activities. This is to ensure that sustainable development is achieved [60]. Most participants highlighted that the role the DCOG plays in the PPP approval process is minimal in practice, although it is empowered by legislation to do more; for instance, to help with any regulatory issues that arise in the implementation of water projects.

## 7. Conclusions

As alluded to in the introductory and conceptual framework sections, there is limited research exploring the interplay among stakeholders and how their perspectives of each other influence the pace of PPP approval. This study specifically provides a deeper understanding of how stakeholders view one another during the PPP approval process, and how this may affect their participation in future municipal PPP projects.

The slow pace of municipal water infrastructure PPP projects in the selected municipalities can be partly explained by tension among stakeholders, as postulated by stakeholder theory. The actors have different objectives driven by self-interest; for instance, the National Treasury's focus is to ensure that the PPP projects are funded in accordance with good financial management principles and the law. Municipal officials (bureaucracy) may have incentives aligned with the requirements of the community; however, councillors are required to approve the implementation of PPP projects. Politicians view PPP projects as long-term projects that are not fully aligned to their political term. Public sector labour unions are generally opposed to the private sector delivering public goods/services, and this impedes the pace of PPP project implementation until commitments regarding job security and conditions of service have been secured.

This paper provides an exposition of how each stakeholder evaluates the (in)effectiveness of other stakeholders during the approval of municipal PPP projects, and how this may influence the adoption of PPPs into the future. The views of respective stakeholders are valuable for PPP practitioners to understand in order to mitigate potential pitfalls and enhance stakeholder cohesion for future PPP projects.

**Author Contributions:** Conceptualisation, T.M. and D.J.F.; methodology, T.M.; validation, D.J.F.; formal analysis, T.M. and D.J.F.; writing—original draft preparation, T.M.; writing—review and editing, T.M. and D.J.F.; supervision, D.J.F.; administration, D.J.F. All authors have read and agreed to the published version of the manuscript.

**Funding:** This research received no external funding.

**Institutional Review Board Statement:** The broader study was approved by the Ethics Committee of the University of Pretoria (EMS187/20 dated 20 November 2020).

**Informed Consent Statement:** Informed consent was obtained from all subjects involved in the study.

**Data Availability Statement:** Data supporting the reported results can be obtained from the corresponding author.

**Conflicts of Interest:** The authors declare no conflict of interest.

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
