# Peer review of "The Role of Stakeholders in the Adoption of Public–Private Partnerships (PPPs) in Municipal Water Infrastructure Projects: A Stakeholder Theory Perspective"

_world, doi:10.3390/world4030026_

Round 1
Reviewer 1 Report (Previous Reviewer 2)
The authors have made improvements in this revision, notably by adding interviewees' demographics. However, I recommend rejection due to persistent issues with the paper's structure and writing, which suggest a need for further academic training.
The manuscript also lacks an analysis section, jumping directly from 'Materials and Methods' to 'Findings'. This omission hampers comprehension and undermines the credibility of the results.
Extensive editing of English language required
Author Response
Thank you so much for the inputs to improve the work

Reviewer 2 Report (Previous Reviewer 1)
Many thanks for addressing all my detailed comments from the last round in a lot of detail.
There are two remaining issues from my side:
Please further clarify your key theoretical contributions in more detail (links to prior studies) in your discussion section.
Please proofread the whole manuscript once more to address typos and spelling mistakes.
Good luck with your revisions.
Please proofread the whole manuscript once more after this round of revisions.
Author Response
We appreciated your inputs to improve the work - Thank you

Reviewer 3 Report (New Reviewer)
This is an interesting article to report on the role of stakeholders in PPP adoption for municipal projects in South Africa. Overall, the article is written very well. However, the authors should address the following points.
· Why is PPP arrangement beneficial for municipal projects from a global perspective?
· How PPP frameworks developed in various countries focusing on the role of stakeholders, see the following work;
o Bing, L., Akintoye, A., Edwards, P. J., & Hardcastle, C. (2005). The allocation of risk in PPP/PFI construction projects in the UK. International Journal of project management, 23(1), 25-35.
o Ahmad, Z., Mubin, S., Masood, R., Ullah, F., & Khalfan, M. (2022). Developing a Performance Evaluation Framework for Public Private Partnership Projects. Buildings, 12(10), 1563.
o Osei-Kyei, R., & Chan, A. P. (2018). A best practice framework for public-private partnership implementation for construction projects in developing countries: A case of Ghana. Benchmarking: An International Journal, 25(8), 2806-2827.
o Jayasuriya, S., Zhang, G., & Yang, R. J. (2020). Exploring the impact of stakeholder management strategies on managing issues in PPP projects. International Journal of Construction Management, 20(6), 666-678.
· What are the parameters or aspects of stakeholder theory used to assess the role of stakeholders? The findings do not reflect any specific evaluation approach. Further, “quotations” from interviewees should be included to support the findings. It seems the author reported the perception of one stakeholder about another but has not covered all from each stakeholder’s perspective. Most importantly, the role perceived by most applicants should be highlighted than with 1-2 responses.
· A discussion section should be included to determine how the role of stakeholder vary and to what extent. How does this influence the success of PPP projects? Most important, what is the significance and practical application of current research? How do you determine the eligibility for PPP projects and the perception of stakeholders’ capability? A wide range of aspects needs to be addressed, such as exposure to PPP, risk-sharing capacity, previous experience with stakeholders, etc.
· Table 1 requires a proper reference.
· Figure 2 has not been explained; what is meant by V1, V2….Vn. Are these perceptions about the value of participation in PPP?
· Few sources, such as [10], have been used repeatedly, indicating a lack of review of relevant articles.
Overall, the quality of the English language is fine.
Author Response
Thank you so much for your inputs - it really added value to our manuscript

Round 2
Reviewer 1 Report (Previous Reviewer 2)
The manuscript has some minor improvements over the previous version. Unfortunately, it fails to reach the standard of World. I insist on my previous reject recommendation, based on several major concerns:
1. The background of the paper is overly lengthy and it is not until page 10 that the authors begin to describe their work.
2. The interview part is quite brief. Demographic information of participants as well as the setting of the interview content is missing. It is even difficult to know whether the interview was actually conducted.
3. It also lacks an analysis section, jumping directly from 'Materials and Methods' to 'Findings'. This omission hampers comprehension and undermines the credibility of the results. I highly recommend adding quantitative analysis on the interview data.
Extensive editing of English language required
Reviewer 3 Report (New Reviewer)
The authors have addressed all the comments.
This manuscript is a resubmission of an earlier submission. The following is a list of the peer review reports and author responses from that submission.
Round 1
Reviewer 1 Report
The paper addresses an interesting and timely research area, investigating the role of stakeholders in public-private partnerships. The authors draw on some prior studies, but a much more critical literature analysis is needed to strengthen the paper’s argument and draw out the (theory) gaps they seek to address. Also, the paper needs to be present much more in-depths findings, stronger discussion and conclusion sections in order to offer value to the reader. Nevertheless, I see significant room for improvement which will help to enhance clarity, readability, practical and theoretical contributions. The following paragraphs address each section of the paper in more detail and provide suggestions on how to revise the paper.
Major concerns:
Introduction:
While the authors establish some links to some extant literature, authors need to establish a more coherent framework for the overall paper. That means, the introduction should clearly indicate the need for this paper in relation to extant research studies. The authors do a good job to explain why the paper is relevant for practice, but miss out to clearly draw out the gap(s) they seek to address with regards to extant studies. Which stakeholders are you referring to, when (in the PPP lifecycle), and why does it matter? It also not clear how this contributes/add to prior studies on PPPs (e.g., Roehrich et al., 2014; Caldwell et al., 2017). This should then help to clarify the key gap(s) the paper’s research question is addressing. I have highlighted some key studies (as a starting point, but please investigate more prior PPP studies) as this will help you to position your argument in prior work.
Conceptual background & Theoretical development:
The authors address some relevant literature, but some more work is needed. This section should offer a more critical reflection of extant studies to draw out clearly where the gaps are. This would help to show how this study seeks to contribute to extant studies. The authors need to more clearly link to extant PPP studies (e.g. James Barlow, Nigel Caldwell, Ilze Kivleniece). For instance, the paper by Kivleniece and Quelin (2012) might prove helpful to link to the ongoing value (co-) creation discussion over the relationship lifecycle. This would help to link to the current risk discussion and how risk may hinder the realization of public and private value in PPPs. The authors need to link clearly the section prior work focusing on the management of PPPs (Zheng et al., 2008). There needs to be a clearer discussion about value (economic and/or social?; bound your key concepts) as this seems to be key in your research study. The authors need to also clearly distinguish between different types of governance mechanisms, e.g. contracts (e.g. performance-based contracts – you may find the special issue on the topic in Industrial Marketing Management (2016) helpful). This would speak not only to the various stakeholders and their values in PPPs, but also how value was created and allocated as well as how stakeholders were managed, thus strengthening your overall argument. A much clearer positioning of the current study (with regards to extant PPP studies) will help to further guide the reader and draw out gap(s) in extant studies.
In part of your manuscript, you talk about ‘bundling’. Please clearly link to prior studies that investigated bundling/unbundling in PPPs (e.g., Roehrich and Caldwell, 2012; Wright et al., 2019).
Methods/Results:
While the others addressed some methodological concerns, some more details are needed to guide the reader:
# Please include a clear rational for your case study (e.g., sampling criteria)?; How was the sector/country selected and what impact do they have on your findings?
# How were the interviewees selected (theoretical sampling?)
# How were data analyzed?; What secondary data did you use and how?
# How were your results verified/checked for consistency/generalisability?
# what are the differences between cases/stakeholders?; How does this differ in terms of different types of value (e.g., economic vs. social)?
Discussions and Conclusions:
Derived from a conceptual background section which did not clearly draw out the gaps the paper seeks to address, the discussion and conclusion sections do offer very little additional value to the reader as it stands. The authors need to offer more fine-grained results here and discuss what they intended to find out in the introduction section (link to research questions(); overall aim of the paper). Overall, the authors need to clearly draw out what the theoretical contributions are and how they add to the existing body of knowledge. This section also needs to clear link back to extant studies to offer some clear value to the reader.
Useful references:
Roehrich et al. (2014). Are public-private partnerships a healthy option? A systematic literature review. Social Science & Medicine, Vol. 113, pp. 110-119.
Wright, S. et al. (2019). Public-Private Partnerships for health services: Construction, protection and rehabilitation of critical healthcare infrastructure in Europe. (pp. 125-151) In: Clark, R. and Hakim, S. (eds.) ‘Public Private Partnerships: Construction, Protection, and Rehabilitation of Critical Infrastructure’, New York: Springer
Reviewer 2 Report
This paper claims to conduct an interview of 31 management officials and claims to use stakeholder theory to find out the roles of stakeholders in PPPs of water infrastructures in South Africa.
Overall, the manuscript in its current form does not meet the standard of World and I recommend reject. Here are my major concerns.
1. Abstract. It is not clear what the authors have done and what their main findings are.
2. Introduction. The paper cites too many papers, which is quite lengthy and verbose.
3. Materials and methods. This paper claims to conduct an interview of 31 management officials. However, it is not clear how the interview is conducted. The content of the interview and demography of participants are missing.
4. Findings. The paper jumps directly to the findings, without any quantitative or qualitative analysis.
5. Extensive editing of English are required.